# A Comprehensive Hydrothermal Co-Liquefaction of Diverse Biowastes for Energy-Dense Biocrude Production: Synergistic and Antagonistic Effects

**DOI:** 10.3390/ijerph191710499

**Published:** 2022-08-23

**Authors:** Guanyu Zhang, Kejie Wang, Quan Liu, Lujia Han, Xuesong Zhang

**Affiliations:** Engineering Laboratory for AgroBiomass Recycling & Valorizing, College of Engineering, China Agricultural University, Beijing 100083, China

**Keywords:** hydrothermal co-liquefaction, synergistic effect, biocrude, energy recovery, sewage sludge, cow manure

## Abstract

Hydrothermal co-liquefaction (co-HTL) is a promising technology to valorize binary or even ternary biowastes into bioenergy. However, the complex biochemical compositions and unclear synergistic effect prevent the development of this technology. Thus, this study explored a comprehensive co-HTL of representative biowastes to investigate the synergistic and antagonistic effects. An apparent synergistic effect on biocrude yield was observed when sewage sludge was co-liquefied with cow manure or wheat straw. Further, the co-HTL of sewage sludge-cow manure was investigated in a detailed manner. The highest yield (21.84 wt%) of biocrude, with a positive synergistic effect (11.37%), the highest energy recovery (47.48%), and a moderate biocrude HHV (34.31 MJ/kg) were achieved from co-HTL at 350 °C for 30 min. Hydrochar and gas products were also characterized to unravel the reaction pathways. Accordingly, this work indicates that sewage sludge co-liquefied with other biowastes can serve as a multi-purpose solution for biowaste treatment and bioenergy production.

## 1. Introduction

Nowadays, the global energy scenario is revolutionizing from petroleum fuel to renewable energy, and biomass energy is a promising alternative to mitigate climate change and develop a sustainable route for bioenergy generation [1]. Thermochemical technologies are effective for valorizing biomass into liquid or gaseous biofuels [2]. Biomass drying/dewatering is an energy-intensive step while implementing the pyrolysis or transesterification process for biofuel production [3]. Of these emerging technologies, hydrothermal liquefaction (HTL) is receiving critical attention because it can valorize wet biomass without any pre-drying steps into valuable products (e.g., biocrude oil, hydrochar, and syngas) at temperatures (280–400 °C) and pressures (10–35 MPa) [4].

To date, numerous researchers have used a variety of biomass feedstocks for biocrude production. However, HTL of individual feedstocks often presents some disadvantages concerning biocrude production for industrialization [5]. For example, individual HTL of low-lipid microalgae resulted in a low biocrude quality, containing abundant N-containing compounds [6]. The HTL of lignin-rich feedstocks should always be operated under more critical conditions, which is not favorable for biocrude production [7,8].

Hydrothermal co-liquefaction (co-HTL) is considered as a more promising technology because using binary or even ternary biomass feedstocks instead of individual biomass feedstock is more practical and economically viable [9]. Moreover, co-HTL of biomass feedstocks can synergistically promote the biocrude yield and quality by means of tuning the biochemical compositions of feedstocks [1,10].

A few studies have implemented the co-HTL of multiple biomass feedstocks. It was established that the co-HTL of microalgae with lignocellulosic biomass at low reaction temperatures (260–320 °C) led to a synergistic effect on biocrude yield [11]. For example, the co-HTL of *Chlorella pyrenoidosa* (CP) with sweet potato waste observed a maximum biocrude conversion of 40.5%, with a high energy value of 35.40 MJ/kg [12]. Furthermore, recent studies have reported that a high biocrude yield and quality were observed while sludge was hydrothermally liquefied as a co-substrate [13,14,15]. For instance, the co-HTL of sewage sludge (SS) with lignocellulosic biomass (67:33 mass ratio) at 340 °C for 20 min achieved a synergistic effect on biocrude production due to the formation of NH_4_^+^ derived from the degradation of protein in SS [16]. Furthermore, it was stated that higher biocrude production was achieved while lignocellulosic biomass was co-liquefied with other feedstocks [17,18,19,20]. As an example, the co-HTL of pine wood with manure/digested SS with a mass ratio of 1:1 at 300 °C for 30 min contributed to the decrease of biocrude viscosity, and a synergistic effect on biocrude production when pine wood was co-liquefied with manure [21].

By contrast, an antagonistic effect during the co-HTL process is sometimes observed. It is proven that the co-HTL of microalgae with lignocellulosic biomass at the higher reaction temperatures (340–360 °C) caused an antagonistic effect on biocrude production [11]. For instance, an obvious antagonistic interaction during the co-HTL of microalgae with organic waste was observed on biocrude production due to attenuated decarboxylation reaction [12]. Because biomass feedstocks usually possess different biochemical compositions, the complexity of biocrude compositions make it challenging to determine the synergistic and antagonistic effects during the co-HTL process [5]. Accordingly, selecting appropriate biomass feedstocks for co-HTL is still an essential concern.

It would be an innovative prospect to investigate the synergistic and antagonistic effects on biocrude yield and quality during the co-HTL of representative biomass categories (i.e., sludge, livestock manure, lignocellulose, and microalgae). Even though several studies have conducted co-HTL of some simple biomass feedstocks, there is no study that has conducted a comprehensive co-HTL process and investigated the synergistic and antagonistic effects among these representative biomass categories on biocrude yield and quality. In particular, the co-HTL of sludge and livestock manure has been rarely studied. The first study within this scope reported co-HTL of sewage sludge with swine manure [22]. It was observed that binary mixture containing ~80% of swine manure and 20% of sewage sludge was smoothly pumpable, and the binary mixture subjected to the co-HTL process gave rise to the highest biocrude yield (42.38 wt%) and maximum synergistic effect, achieving the best HHV of 36 MJ/kg.

With these premises, the present work aimed to expand the state of knowledge and critically assess the technical feasibility and efficiency of the co-HTL of representative biomass feedstocks (i.e., sewage sludge, cow manure, wheat straw, and *Chlorella pyrenoidosa*) focusing on the biocrude yield and quality, as well as the synergistic effect.

To the best of our knowledge, this research is the first study that has explored the co-HTL of binary sewage sludge-cow manure, cow manure-wheat straw, and cow manure-microalgae, etc. The effect of the main reaction parameter on the biocrude yield and quality during the co-HTL of optimized binary biomass feedstocks was further explored. Overall, this study showed significant implications in understanding the synergistic and antagonistic effects during the co-HTL of the representative biomass categories and provides a new sight into the enhancement of biocrude yield and quality.

## 2. Materials and Methods

### 2.1. Materials

Cow manure (CM) was supplied by a local dairy farm (Treasure Island farm in Beijing, China). Sewage sludge (SS) was provided by a municipal wastewater treatment plant of Beipai Group in Beijing, China. Wheat straw (WS) was obtained from Shangzhuang experimental station of China Agricultural University in Beijing, China. *Chlorella pyrenoidosa* (CP) was purchased from Guangyu Biotechnology Co., Ltd. (Shanghai, China). Initially, all raw biomass materials were ground into 20–40 mesh by means of a high-speed grinder (RT-34, Taiwan Rongcong Precision Technology Co., Taizhong, Taiwan, dried at 105 °C for 24 h, and stored at −20 °C for future use. All chemicals and reagents used in the experiment, including methanol (>99.5%) and dichloromethane (DCM, >99.5%), were purchased from Beijing Lanyi Chemical Co., Ltd. (Beijing, China).

### 2.2. HTL Experimental Procedures

The individual HTL and co-HTL experiments were carried out in the stainless batch reactor (about 10 mL of internal volume) assembled with a ½ inch and a ¼ inch Swagelok cap, plus a 316 ss ½ inch Swagelok port connector. The reactor was preloaded with 0.5 g of individual feedstock or binary mixture (with the fixed mass ratio of 1:1) and 10 g of deionized (DI) water for each test. Prior to the tests, the assembled reactor was vented with N_2_ to get rid of the air present in the reactor. For safety considerations, the reactor was immobilized in a self-assembly protective device. A detailed diagram of this apparatus is shown in Figure 1. It was evidenced that the reaction temperature kept at 350 °C was optimal for the co-HTL of biomass wastes to achieve a high biocrude yield and a higher HHV [23,24,25,26]. Thus, the reaction temperature for both individual and co-HTL tests were maintained at 350 °C in this study. After the muffle furnace equilibrated at 350 °C for 30 min, the reactor, along with the self-assembly protective device, were rapidly placed in a preheated furnace to proceed with the experiments. As the set-point residence time was reached, the reactor, together with the protective device, were quickly taken out from the muffle furnace and quenched in a water bath for 20 min. Eventually, the reactor was separated from the protective device and the water remaining on the external surface of the reactor was thoroughly wiped.

### 2.3. Product Recovery

After the tests, the reactor was slowly depressurized to discharge the gas. The effluent was poured out and dissolved into a 30 mL DCM, then, the interior of the reactor and the caps were washed multiple times with DCM (10 mL) and DI water (10 mL) in small aliquots. Afterwards, the insoluble solid phase (hydrochar) was separated by filtration from the mixture. The DCM-soluble organic phase is termed as biocrude or biocrude oil (BC), whilst the water-soluble phase is called the aqueous phase. The hydrochar (HC) and aqueous phase (AQ) were dried at 60 °C overnight. At the same time, the biocrude oil was recovered by flowing air over the DCM-soluble phase at room temperature until the weight change was less than 0.1 mg. The as-received biocrude and hydrochar are denoted as BC-*X-Y* and HC-*X-Y*, respectively, where *X* represents the feedstock (including individual feedstocks and binary feedstocks) and *Y* is the residence time in min (i.e., 15, 30, 45, 50, 60, 90, and 120). The gravimetric yield of the biocrude, hydrochar, and aqueous phase was measured as the weight of the dried product divided by the weight of the individual feedstock or binary feedstocks loaded into the reactor. The gravimetric yield of gas and volatiles was determined by difference. All the tests were implemented in triplicate, and the product yields were determined as the mean values of the tests.

### 2.4. Analytical Methods

The CHNS elemental analysis of the biomass feedstocks, biocrude oils, and hydrochar were conducted by an elemental analyzer (Elementar Vario ELIII, Hanau, Germany). The content of moisture, volatile matter, and ash were measured by an automatic industrial analyzer (YXGYFX 7705B, U-Therm, Changsha, China). The amount of fixed carbon was calculated by the subtraction method. Thermogravimetric analysis (TGA) of biomass feedstocks and hydrochars was performed by a thermogravimetric analyzer (SDTQ600, TA Instruments, Newcastle, WA, USA). The functional groups of biomass feedstocks, biocrude oils, and hydrochars were evaluated by Fourier transform infrared spectroscopy (FTIR, spectrum 400, PerkinElmer, Waltham, MA, USA).

The gas was collected by a 10 mL syringe and then characterized offline by a Shimadzu GC-2014C gas chromatographer (GC, Shimadzu Corp., Kyoto, Japan) equipped with a thermal conductivity detector (TCD). A standard gas mixture containing H_2_, CO, CH_4_, CO_2_, C_2_H_4_, and C_2_H_6_ was quantified to calibrate the proportion (vol.%) of gaseous fractions. The gas compositions (≥C_3_) were not observed or negligible in this study. All GC measurements were repeated to ensure reproducibility.

### 2.5. Data Analysis

As mentioned above, the mass of gas and volatiles (GV) was denoted as the following formula (Equation (1)):(1)$$Mass_{GV}=Mass_{Feedstock}-Mass_{BC}-Mass_{HC}-Mass_{AQ}\;$$

Yields of the product distributions were determined using Equation (2).
(2)$$Y_{i}(wt\%)=\frac{mass_{i}}{mass_{Feedstock}}\times 100\;$$
where *i* is the biocrude oil, hydrochar, aqueous phase, and gas and volatiles from each experiment.

In this study, the synergistic effect (*SE*), as a crucial co-HTL evaluation index, was expressed as the following formulas, Equations (3) and (4)):(3)$$Y_{Theoretical}=Y_{Feedstock1}\times X_{Feedstock1}+Y_{Feedstock2}\times X_{Feedstock2}$$
(4)$$SE=\frac{\;Y_{Experimental}-Y_{Theoretical}}{Y_{Theoretical}}\times 100$$
where $Y_{Theoretical}$ and $Y_{Experimental}$ are the theoretical and experimental values from the co-HTL process, respectively; $Y_{Feedstock1}$ and $Y_{Feedstock2}$ represent the values from HTL of pure individual biomass feedstock 1 and 2, respectively; and $X_{Feedstock1}\;$and $X_{Feedstock2}\;$are the mass contents (50 wt%) of two feedstocks participating in co-HTL, respectively.

The higher heating value (*HHV*) of the biomass feedstocks, biocrude oils, and hydrochars were calculated based on the elemental composition through Dulong’s formula (Equation (5)) [27].
(5)HHV(MJ/kg)=0.3414C+1.4445H−(N+O−1)/8
where *C*, *H*, *O*, and *N* represent the carbon, hydrogen, oxygen, and nitrogen contents, respectively.

The feedstock conversion rate (*CR*) was determined by subtracting hydrochar yield ($Y_{HC}$) from the cumulative yields of products, as shown in Equation (6).
(6)$$CR\;\left(\%\right)=100-Y_{HC}$$

The energy recovery (*ER*) is calculated by using Equation (7) [28]:(7)$$ER\;\left(\%\right)=\frac{\;HHV_{BC}\times Y_{BC}}{HHV_{Feedstock}}\times 100$$
where $HHV_{BC}$ and $HHV_{Feedstock}\;$are the energy value of biocrude and feedstock, respectively, and $Y_{BC}$ represent the biocrude yield.

## 3. Results and Discussion

### 3.1. Feedstock Characterization

To understand the synergistic effect on biocrude yield and quality during the co-HTL of different binary biomass mixtures, the four representative biomass feedstocks were initially characterized by proximate and ultimate analysis, as given in Table 1. The proximate analysis showed that CP had a higher content (73.90%) of volatile matter than other feedstocks, which was attributed to the presence of carbohydrates, lipids, fatty acids, and proteins. By contrast, a lowest content (44.26%) of volatile matter was found for SS, which was due to the high share (44.33%) of ash in SS. It was reported that ash in SS was mainly composed of mineral contents, which played an important role during the HTL process [15]. The ultimate analysis of the four feedstocks showed that CP presented a higher C (48.47%) and H (10.58%) contents than the others, which was related to the higher amounts of carbohydrates in CP. On the contrary, the lowest C content (30.33%) was observed for SS due to the high ash content. WS as a typical agricultural biomass possessed the lowest H content (3.56%) and N content (0.88%), but the highest O content (39.23%), which resulted in a low HHV (15.80 MJ/kg). It was discerned that the higher N content (8.58%) in CP possibly originated from the crude proteins. The highest S content (1.24%) was detected in SS, which was probably ascribed to some amino acids and proteins [29]. CM had a relatively low S content (0.22%) and N content (1.10%); the low contents of heteroatoms in CM make it promising as a fuel precursor.

As also demonstrated in Table 1, the elemental composition indicated that CP had a higher H/C ratio, which could promote the biocrude production rate via the HTL process [30]. Moreover, CP revealed a relatively low O/C ratio (0.42) but a high N/C ratio (0.15). These results could reflect the highest HHV (27.54 MJ/kg) of CP. Although SS also showed comparable H/C, O/C, and N/C ratios, the highest ash content led to the lowest HHV, ca. 14.66 MJ/kg. In contrast, the lowest H/C ratio (0.93) and N/C ratio (0.02) but the O/C ratio (0.65) were obtained for WS, affirming the low HHV. Thus, the characterization of the representative biomass feedstocks suggested the high liquefaction potential, and the co-HTL of these biomass feedstocks would considerably affect biocrude production and quality.

### 3.2. Hydrothermal Co-Liquefaction of Four Representative Feedstocks: Investigating Synergistic and Antagonistic Effects

To well determine the synergistic effect and antagonistic effects of these representative biomass feedstocks on biocrude yield and quality, the reaction condition (reaction temperature, 350 °C; residence time, 30 min) was constantly maintained for both individual HTL and co-HTL processes. The product yield distributions and conversion rates from individual HTL and co-HTL of four representative biomass feedstocks are illustrated in Figure 2. As regards to individual HTL of four biomass feedstocks (Figure 2a), the highest biocrude yield (33.38 wt%) was observed from the HTL of individual CP, which in turn gained the lowest yield (3.87 wt%) of hydrochar due to the high conversion rate (96.13%) of volatile matter. The higher biocrude yield from the individual HTL of CP was attributed to the abundant amounts of carbohydrates, lipids, and proteins in CP, which were reported to be more favorably converted than lignocellulose [31,32]. It was noted that the biocrude yield was observed to be inversely correlated with ash [33]; SS (44.33 wt%), CM (17.52 wt%), and WS (10.44%) presented higher ash content than CP (4.69 wt%). Thus, the highest hydrochar yield (51.99 wt%) but the lowest biocrude yield (18.00 wt%) were obtained from the individual HTL of SS. For comparison, Vardon et al. reported that a little lower biocrude yield (32. 6 wt%) was achieved from HTL of *Spirulina* algae at 300 °C with the residence time of 30 min [34], and Obeid et al. also achieved a comparable biocrude yield (30 wt%) from HTL of *Tetraselmis* sp. microalgae at 350 °C, with a residence time of 20 min [35]. It should be noted that Saba et al. observed much lower biocrude yields from the HTL of sludge and manure at 300 °C with 30 min (sludge: about 12 wt% vs. 18.00 wt%; manure: nearly 12 wt% vs. 21.22 wt%) [21]. Thus, it was concluded that the reaction condition (reaction temperature of 350 °C and residence time of 30 min) in our study was demonstrated to contributing to a high biocrude yield from the HTL process.

The product yield distributions and conversion rate from the co-HTL of binary biomass feedstocks are depicted in Figure 2b. The biocrude yield from the co-HTL of these binary feedstocks did not present great differences. The co-HTL of binary CP/WS revealed a little higher biocrude yield (26.36 wt%) than the co-HTL of other binary mixtures, even though the maximum conversion rate, ca. 86.69%, was observed. This was because most of the volatile matter was converted into gas and volatiles, with a high yield of 43.61 wt%. The co-HTL of binary CM/WS gave rise to the lowest biocrude yield (21.49 wt%). Apparently, when SS was co-liquefied with other biomass feedstocks, the high hydrochar yield was obtained, maximized at 40.30 wt% from the co-HTL of binary SS/CM. This was due to the fact that SS and CM showed a higher ash content than the other two feedstocks. Importantly, when CP was hydrothermally co-liquefied with other feedstocks, the process was prone to generating more gas and volatiles, gaining a highest yield of 43.97 wt% from co-HTL of binary CP/CM.

During the co-HTL process, the biomass biochemical compositions (carbohydrates, lignin, cellulose, hemicellulose, lipids, and proteins) significantly affected the biocrude production [36,37]. To determine the synergistic effect on biocrude production, the theoretical biocrude yield from the co-HTL process was calculated according to Equation (3). The synergistic effect on the biocrude yield from the co-HTL of different binary biomass feedstocks is depicted in Figure 3. As for the co-HTL operations of binary SS/CM and SS/WS, a positive synergistic effect was found on biocrude yield, reaching 11.37% and 20.37%, respectively. The positive synergistic effect on biocrude yield was ascribed to the presence of alkaline and alkaline earth metals (e.g., K, Na, and Ca) in SS and the N-containing composition producing ammonia, serving as a base catalyst for the degradation of biomass feedstocks [14,38]. As similar outcome was also reported by another study, where a maximum synergistic effect was found during the co-HTL of SS with rice straw at a mixing ratio of 1:1, and the mineral contents in SS played an important role in catalyzing the reactive substances of rice straw [15]. On the other hand, an obvious antagonistic effect on biocrude yield was found when CP was hydrothermal co-liquefied with other biomass feedstocks, which was in line with result reporting that the high reaction temperatures (340–360 °C) led to an antagonistic effect on biocrude production [11].

Typically, the elemental analysis, the atomic molar ratio, HHV, and energy recovery are measured to evaluate the biocrude quality and energy density [26]; the characteristics of biocrude oils from the co-HTL of different binary mixtures are summarized in Table 2. Broadly speaking, the biocrude oils produced from CP co-liquefied with other feedstocks showed a higher C content (over 72 wt%) than the other binary mixtures, possibly due to the high volatile and C contents in CP. By contrast, the biocrude oils from CP co-liquefied with other feedstocks presented the lower O content (approximately 14 wt%). Interestingly, it was found that when SS was co-liquefied with other feedstocks, the higher H content (over 8 wt%) in the biocrude oils were obtained; furthermore, these biocrude oils revealed a higher H/C molar ratio (around 1.40).

Based on the elemental compositions of the biocrude oils, the HHV of these biocrude oils was determined, and is also listed in Table 2. Notably, the biocrude oils gained from the co-HTL of binary SS/CM and SS/CP exhibited a higher HHV than those from the other binary mixtures, reaching 34.31 MJ/kg and 35.16 MJ/kg, respectively. Furthermore, the energy recovery calculated by combining biocrude yield and HHV results offers a vital overview of the evaluation of the co-HTL of diverse binary mixtures, as explained in Table 2. It is noteworthy that the energy recovery from the co-HTL of binary SS/CM and SS/WS was 47.48% and 53.43%, respectively, which was much higher than that garnered from the co-HTL of other binary mixtures. Taking the biocrude yield, synergistic effect, biocrude quality, and energy recovery into consideration, SS/CM was the optimal binary mixture for the co-HTL process to achieve a high biocrude yield, synergistic effect, HHV, and energy recovery. Additionally, the co-HTL of sludge with manure has rarely been studied [22], and the co-HTL of binary SS/CM has never been conducted for biocrude production. Thus, it is necessary to investigate the co-HTL of binary SS/CM in detail to unravel experimental investigations and to optimize biocrude yield and quality. As a result, it was concluded that binary SS/CM for co-HTL was more appropriate to produce biocrude, and the binary SS/CM would be used as the optimal mixture to further optimize the biocrude yield and quality.

### 3.3. Hydrothermal Co-Liquefaction of Sewage Sludge-Cow Manure: Optimizing Biocrude Yield and Quality

#### 3.3.1. Product Yields of Co-HTL of Binary SS/CM

Concerning that the product yields strongly depend upon the processing parameters, the influence of residence time in product yields and conversion rate was investigated, as shown in Figure 4. Regarding the individual HTL of SS (Figure 4a), the increase of residence time from 15 min to 90 min contributed to the gradual increase of biocrude yield, gas and volatiles yield, and conversion rate, which were kept stable by further increasing the residence time to 120 min. As for the individual HTL of CM (Figure 4b), an obvious increase in biocrude yield was obtained when the residence time was prolonged from 15 min to 30 min. However, as the residence time continued to increase, a slight drop in biocrude yield was found. The increase of residence time was conducive to the increase of the yield of gas and volatiles from 20.98 wt% to 48.26 wt%. The conversion rate from the individual HTL of CM increased as the residence time increased from 15 min to 60 min, and then fluctuated at ~75% in the residence time range of 60–120 min.

The biocrude yield from the co-HTL of binary SS/CM experienced a comparable tendency with the biocrude yield from the individual HTL of CM, as shown in Figure 4c. Specifically, the biocrude yield was elevated from 11.58 wt% to 21.84 wt% when the residence was prolonged from 15 min to 30 min, while the biocrude exhibited a slight decrease to 19.32 wt% as the residence was further augmented to 120 min. The yield of gas and volatiles revealed an upward tendency from 20.98 wt% to 48.26% by prolonging the residence time from 15 min to 120 min. The conversion rate was significantly improved from 43.49% to 59.70% as the residence was elevated from 15 min to 30 min, and it exhibited a slight increasing trend from 59.70% to 63.34% with the increase of residence time from 30 min to 120 min. It has also been summarized that, during biomass liquefaction, biocrude yield can be maximized before decreasing for very long residence times, whilst gas yield and biomass conversion rate would continuously go up before the saturation point [39]. The hydrochar yield from either individual HTL or co-HTL processes continuously declined with the increase of residence time.

For the co-HTL tests, the experimental biocrude yield was compared with the theoretical biocrude yield to evaluate the synergistic and antagonistic effects. As illustrated in Figure 5, it was noticed that there was an obvious synergistic effect on biocrude yield in the residence time range of 15–120 min. Although the optimal synergistic effect (13.17%) was obtained from the co-HTL process operated for 15 min, the practical biocrude yield was extremely low (11.58 wt%), which was not economically feasible for scale-up. The residence time set at 30 min for the co-HTL of binary SS/CM not only contributed to a decent synergistic effect (11.37%) on biocrude yield but also observed a maximum biocrude yield (21.84 wt%). Accordingly, the residence time of 30 min was the most appropriate parameter for the co-HTL of binary SS/CM in consideration of biocrude yield and the synergistic effect.

#### 3.3.2. Characteristics of Biocrude Oils

The characteristics of biocrude oils from the individual HTL of SS or CM and co-HTL of binary SS/CM by operating varied residence times (15–120 min) is presented in Table 3. In general, the C and H contents of biocrude oils were in the range of 69.62–76.41 wt% and 7.01–10.22 wt%, respectively. There was higher C and H contents in biocrude oils than in corresponding biomass feedstocks, affirming the energy accumulation during the individual HTL and co-HTL process. As for the characteristics of biocrude oils produced from the individual HTL of SS or CM, the different biochemical compositions of SS and CM reflected the difference in biocrude characteristics. The high level of O was measured in the biocrude oil from the individual HTL of CM (21.55 wt% in BC-CM-30 vs. 10.73 wt% in BC-SS-30, see Table 3), whose elemental analysis exhibited a higher O mass fraction in comparison with SS (39.23 wt% in CM vs. 15.93 wt% in SS, see Table 1). A lower N mass fraction was found in the biocrude oil from the individual HTL of CM (1.54 wt% in BC-CM-30 vs. 3.04 wt% in BC-SS-30), which contained less N than SS (1.10 wt% in CM vs. 3.60 wt% in SS). Similar results were also reported in other study [32].

Notably, the increase of residence time from 15 min to 30 min did not show the obvious variations in C content; with the residence time further increasing to 120 min, the C content was gradually improved. Among these biocrude oils produced from the co-HTL of binary SS/CM, BC-SS/CM-90 and BC-SS/CM-120 presented the higher shares of C (>75 wt%) and H (~9 wt%), suggesting a higher energy conversion rate during the co-HTL process. The amount of O in biocrude oils was comparatively reduced as compared to that in binary SS/CM, implying that the deoxygenation reaction occurred during the co-HTL process. Furthermore, the increase of residence time from 15 min to 30 min did not strongly affect the O content, while by further augmenting the residence time to 120 min, the O content was gradually lowered to 10.75%. In addition, S content revealed a comparable trend with O content, while the N content in the biocrude oils was in the range of 1.94–3.07%.

For the HHV of biocrude oils from the individual HTL process, BC-SS-30 showed a higher HHV (38.47 MJ/kg) than BC-CM-30 (31.13 MJ/kg), which was ascribed to the lower O content in BC-SS-30. Regarding the HHV of biocrude oils from the co-HTL process, the highest HHV (37.67 MJ/kg) was found for BC-SS/CM-120, suggesting that the prolonged residence time possibly contributed to the deoxygenation and decarboxylation reactions and thus resulted in the higher energy density. For comparison, the HHV of the biocrude oils from the co-HTL of binary SS/CM in this study was more than twice that of the HHV (e.g., 17.8 MJ/kg) of biocrude oils gained from the co-HTL of mixed-culture algae with swine manure [41].

The energy recovery considering both biocrude yield and composition provides a more apparent overview on the evaluation of the co-HTL of binary SS/CM. As explained in Table 3, most of the biomass energy was captured in biocrude oils through the co-HTL process because the energy recovery of biocrude oils from the co-HTL process with the residence time of 30–120 min was over 42%. Importantly, prolonging the residence time did contribute to the increase of energy recovery. The BC-SS/CM-30 revealed the highest biocrude yield (21.84 wt%) and energy recovery (47.48%), with a moderate HHV of 34.31 MJ/kg. Moreover, the practical energy recovery of BC-SS/CM-30 was larger than the theoretical energy recovery (42.63%), achieving a synergistic effect of 11.38% (Appendix A). By contrast, the lowest biocrude yield (11.58 wt%) and energy recovery (26.04%) was observed for BC-SS/CM-15 due to the incomplete conversion of binary feedstocks in the short duration time (15 min). These observations suggested that the biocrude yield exhibited a more significant impact at the energy recovery than the variations in residence time.

The quality of the biocrude oils in terms of H/C and O/C atomic ratios is demonstrated in a Van Krevelen diagram (Figure 6). The O/C ratios of the biocrude oils (0.11–0.23) were dramatically lowered as compared to raw SS (0.39) and CM (0.64) because of the decarboxylation and dehydration reactions taking place during the individual HTL and co-HTL processes, contributing to energy-dense biocrude production. The H/C and O/C atomic ratios from the co-HTL process lay in the area between biocrudes produced from individual HTL of SS or CM, suggesting that their quality reflected the biochemical compositions of both SS- and CM-derived biocrude oils. Similar phenomenon during the co-HTL of sludge with lignocellulosic biomass was also observed by other studies [10,13]. The maximum HHV of biocrude from the co-HTL process was 37.67 MJ/kg, with a high H/C atomic ratio of 1.42 and lowest O/C ratio of 0.11. For comparison, the data regarding Petro-crude is also given in Table 3 and Figure 6. The O/C ratio (>0.1) of the biocrude oils did not satisfy the Petro-crude specification [40]. These outcomes suggest that all biocrude oils need to be upgraded for O reduction.

FTIR analysis was used to determine the functional groups, vibration mode, and spectra strength of biomass biocrude oils, as demonstrated in Appendix A. It was notably found that there were almost similar patterns of spectra among these biocrude oils. Specifically, the spectra in the range of 2800–3000 cm^−1^ and 1350–1460 cm^−1^ were assigned to strong peaks of the C-H stretching (-CH_3_ and -CH_2_ vibrations) of hydrocarbons, demonstrating the decarboxylation reaction during the individual HTL and co-HTL processes [10,28]. The decarboxylation and dehydration reactions occurring in the process improved the removal of oxygen in the form of CO_2_ [4], which was consolidated by the dominant CO_2_ proportion in the gases (Appendix A). The bands at 1200–1300 cm^−1^ and 1020–1100 cm^−1^ were attributed to the O-H stretching and C-O stretching from phenols and alcohols in biocrude oils [10]. The absorbance in the range of 1590–1800 cm^−1^ represented C=O vibration, indicating the presence of carboxylic acids, ketones, aldehydes, and esters in biocrude oils [42]. Apparently, BC-SS/CM-15 revealed the highest absorbance at 1590–1800 cm^−1^, indicating the significant presence of carboxylic acids, ketones, etc. The medium peak at 750–800 cm^−1^ was related to aromatic compounds in biocrude oils [28]. Comparably, the similar functional groups including C-O, O-H, C-H, C=O, and N-H were found in other co-HTL studies reported elsewhere [3,14].

#### 3.3.3. Hydrochar Characterization

Elemental analysis of hydrochar samples was conducted by excluding the ash content from the hydrochars. A higher ash content was detected for HC-SS-30, while the higher C fraction (40.94 wt%) was shifted in HC-CM-30, as listed in Table 4. Moreover, HC-CM-30 also observed a higher N fraction (2.01 wt%), indicating the migration of N in hydrochar through repolymerization and cyclization reactions of N-containing compounds [22]. The H/C atomic ratio is utilized to represent the aromaticity of hydrochars [13]. The low H/C ratio (<2) of the hydrochars suggested that the majority of H content was transferred into the biocrude oils. Notably, the prolonged residence time was beneficial for the increase of aromaticity of hydrochars from the co-HTL process. The low HHV of the hydrochars also validated the main conversion route of individual or binary feedstocks into biocrude, aqueous phase, gas, and volatiles by decarboxylation and dehydration reactions.

The functional groups on hydrochar were identified by FTIR analysis (Appendix A). The peaks of FTIR spectra at approximately 1035 cm^−1^, 1545 cm^−1^, 1680 cm^−1^, 2900 cm^−1^, and 3400 cm^−1^ were assigned to the C-O, C=C, C=O, C-H, and O-H/N-H vibrations in the hydrochars, which were comparable with those of raw SS and CM. Thus, the presence of C-H, C=O, O-H, and COOH surface functional groups in hydrochar possibly allows it to act as a promising adsorbent for wastewater treatment [10].

The thermal degradation behaviors of SS, CM, and hydrochars were analyzed by TGA, as shown in Appendix A. The thermal degradation of SS and CM mainly occurred at 150–400 °C (see Appendix A), which was due to the devolatilization, and most organic matters were degraded in the temperature range. The DTG curves (Appendix A) of SS and CM peaked at approximately 280 °C and 320 °C, respectively. These outcomes indicated that SS was more favorable than CM for biocrude production due to the low-severity degradation. The higher rate of CM suggested the higher organic content in CM. Obviously, the TG analysis of hydrochars showed that HC-SS/CM-15 exhibited a considerable loss (over 30%), indicating the low thermal stability. The primary reason for this was that the short residence time of 15 min cannot allow the binary SS/CM feedstocks to undergo a thorough degradation. Hydrochars produced with prolonged residence time (≥30 min) indicated much higher thermal stability, which was thermally stable up to 400 °C, and only ~10% of loss in the range of 400–800 °C was found.

#### 3.3.4. Analysis of Gaseous Products

The gas and volatiles were the major byproduct from both the individual HTL and the co-HTL processes, with the yield varying from 13.24 wt% to 48.26 wt%. As regards the co-HTL process, the yield of gas and volatiles gradually increased from 17.39 wt% to 38.99 wt% as the residence time was prolonged from 15 min to 120 min during the co-HTL of binary SS/CM. This is possibly explained by the fact that the augment of residence time could prolong the reactions during the co-HTL process so that more gas and volatiles should be generated. The gaseous compositions were identified and quantified by GC, which would be beneficial for the demonstration of the reaction pathways during the process. It was observed that the gases mainly consisted of CO_2_, CO, H_2_, and CH_4_; trace amounts of C_2_H_4_ and C_2_H_6_ were also found. The results regarding the HTL gas phase are presented in Appendix A. In Section 3.3.2, it was inferred that the oxygen removal was strongly related to decarboxylation reactions during the individual HTL and co-HTL processes. Because CO_2_ was dominant in the gases, maximizing at 73.40 vol%, it reaffirmed that the decarboxylation reaction by expelling O in the form of CO_2_ predominantly took place during the co-HTL process. It was also discerned that the increasing tendency of CO_2_ volumetric concentration from 41.07 vol% to over 73 vol% was observed with the residence prolonged from 15 min to 120 min, which resulted from the enhanced decarboxylation extent as the residence time increased.

## 4. Conclusions

To summarize, this study explored a comprehensive co-HTL of representative biomass feedstocks to investigate the synergistic and antagonistic effects on biocrude yield and quality. The obvious synergistic effect on biocrude yield was observed from co-HTL of binary SS/CM and SS/WS. The highest biocrude yield (21.84 wt%) with a positive synergistic effect (11.37%), the highest energy recovery (47.48%), and a moderate biocrude HHV (34.31 MJ/kg) was gained from the co-HTL of binary SS/CM at 350 °C for 30 min. From the perspective of obtaining energy-dense biocrude, SS co-liquefied with other biowastes acts as a multi-purpose solution for biowaste treatment and bioenergy production.

## Figures and Tables

**Figure 1 ijerph-19-10499-f001:**
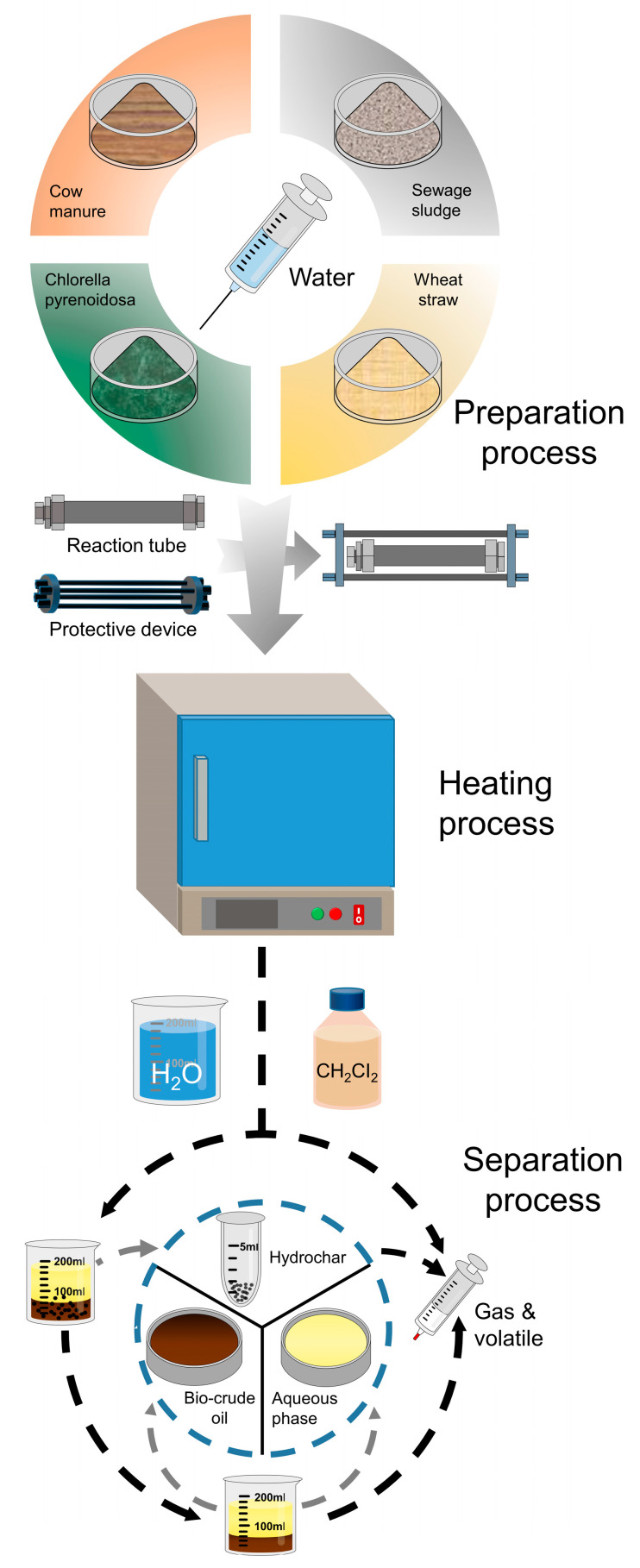
Schematic diagram of the hydrothermal co-liquefaction procedures integrated with product recovery.

**Figure 2 ijerph-19-10499-f002:**
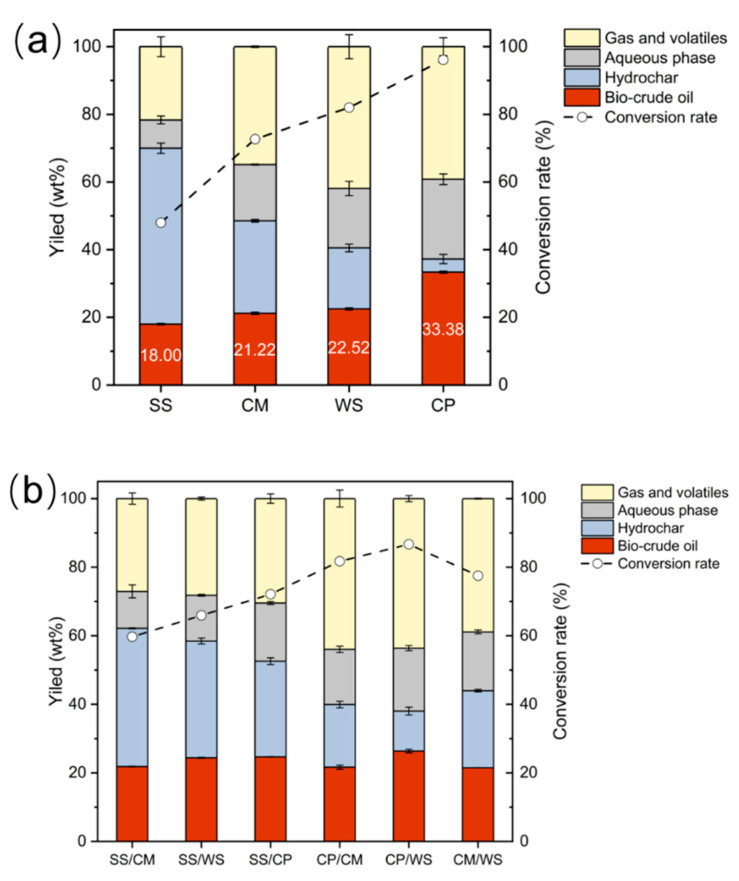
Product yield distributions and conversion rate from HTL of (**a**) individual feedstocks and (**b**) binary feedstocks. Reaction condition: reaction temperature, 350 °C; residence time, 30 min; binary mass ratio, 1:1.

**Figure 3 ijerph-19-10499-f003:**
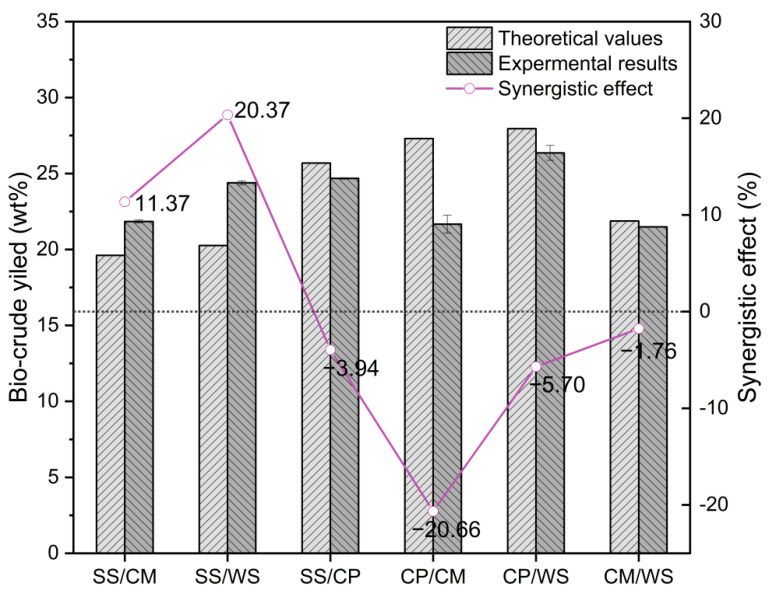
Synergistic effect on the biocrude yield from co-HTL of different binary biomass feedstocks. Reaction condition: reaction temperature, 350 °C; residence time, 30 min; binary mass ratio, 1:1.

**Figure 4 ijerph-19-10499-f004:**
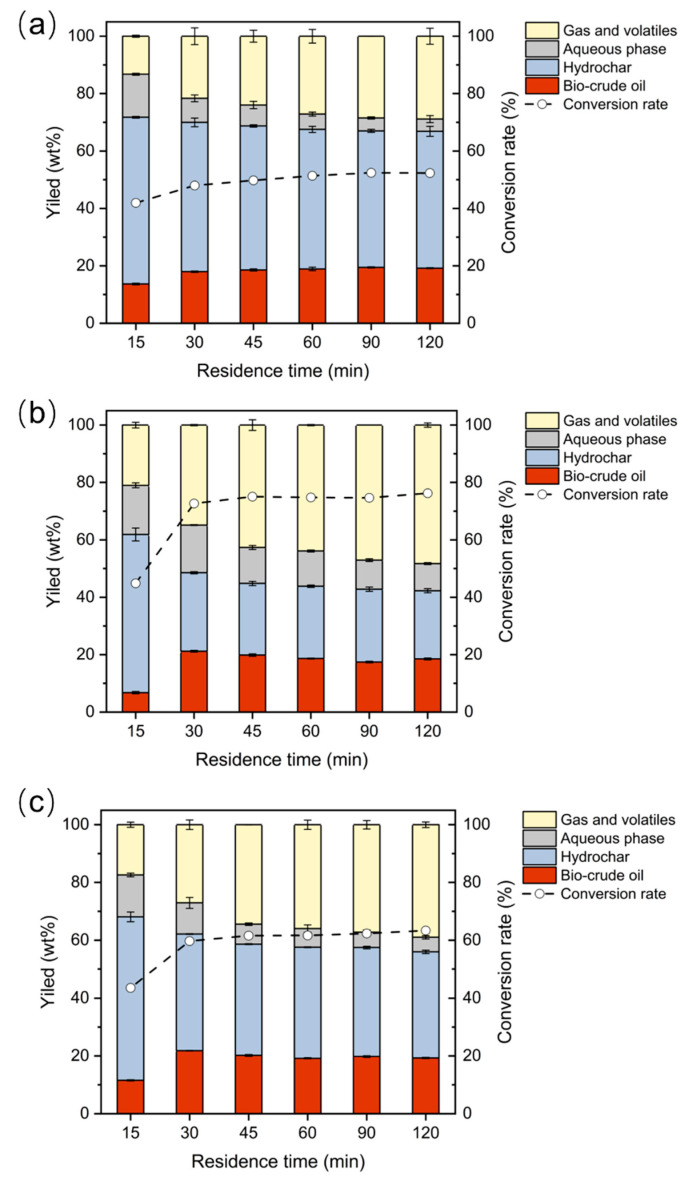
The effect of residence time on product yield distributions and conversion rate from the HTL of (**a**) SS, (**b**) CM, and (**c**) binary SS/CM at 350 °C.

**Figure 5 ijerph-19-10499-f005:**
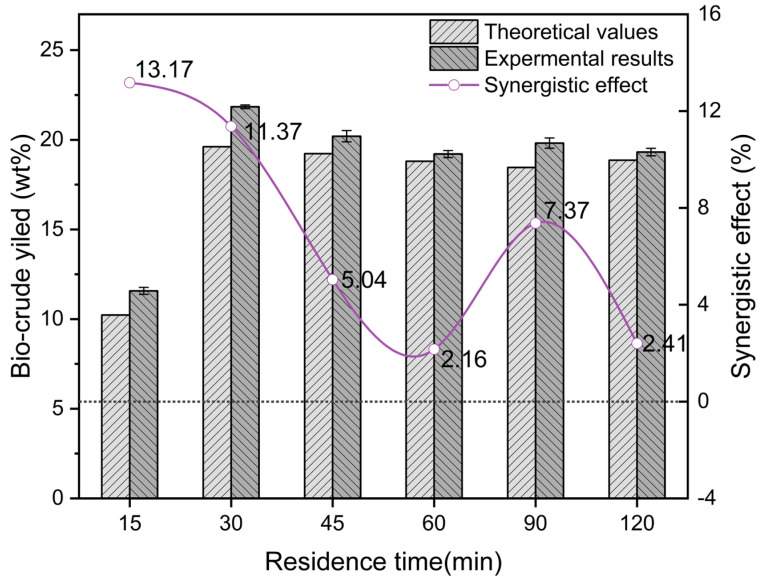
Synergistic effect on the biocrude yield from co-HTL of binary SS/CM as a function of residence time.

**Figure 6 ijerph-19-10499-f006:**
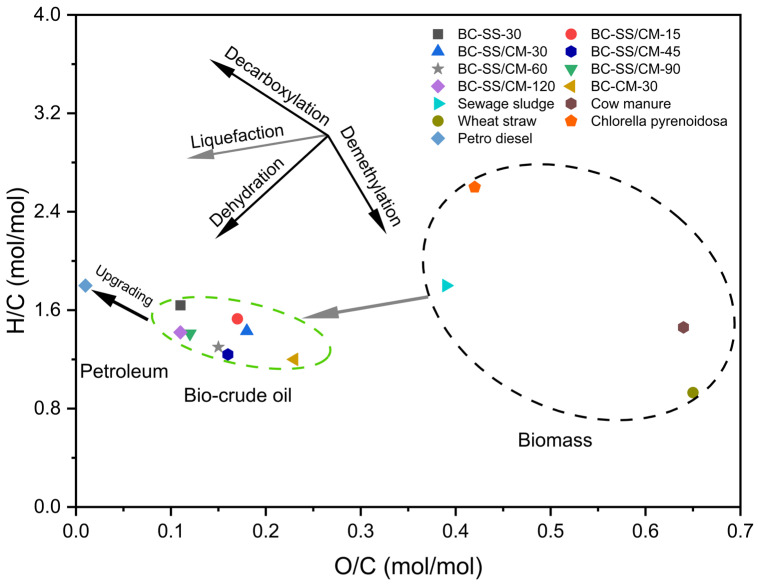
Van Krevelen diagram of biomass feedstocks, biocrude oils, and Petro-crude [40].

**Table 1 ijerph-19-10499-t001:** Proximate and ultimate analysis of four representative biomass feedstocks.

Characteristics	Sewage Sludge (SS)	Cow Manure (CM)	Wheat Straw (WS)	*Chlorella pyrenoidosa* (CP)
*Proximate analysis (wt%)*				
Moisture	4.44	3.90	4.81	5.24
Volatile matter	44.26	61.37	70.02	73.90
Fixed carbon ^a^	6.98	17.22	14.74	16.19
Ash	44.33	17.52	10.44	4.69
*Ultimate analysis (wt%)*				
C	30.33	41.07	45.53	48.47
H	4.58	5.04	3.56	10.58
N	3.60	1.10	0.88	8.58
S	1.24	0.22	0.36	0.87
O ^a^	15.93	35.05	39.23	26.81
H/C (mol/mol)	1.80	1.46	0.93	2.60
O/C (mol/mol)	0.39	0.64	0.65	0.42
N/C (mol/mol)	0.10	0.02	0.02	0.15
HHV(MJ/kg)	14.66	16.91	15.80	27.54

^a^ Determined by difference, O = 100 − sum of (C, H, N, S, and Ash), Fixed carbon (%) = 100 − sum of (Moisture, Volatile matter, and Ash).

**Table 2 ijerph-19-10499-t002:** Characteristics of biocrude oils from the co-HTL of different binary biomass feedstocks. Reaction condition: reaction temperature, 350 °C; residence time, 30 min; binary mass ratio, 1:1.

Characteristics	BC-SS/CM	BC-SS/WS	BC-SS/CP	BC-CP/CM	BC-CP/WS	BC-CM/WS
C (wt%)	71.04	70.57	72.45	72.72	72.67	71.46
H (wt%)	8.54	8.09	8.65	7.45	7.43	6.41
O ^a^ (wt%)	16.64	17.95	12.89	14.59	14.67	20.75
N (wt%)	2.60	2.44	4.70	4.78	4.92	1.24
S (wt%)	1.18	0.95	1.31	0.46	0.31	0.14
H/C (mol/mol)	1.43	1.37	1.42	1.22	1.22	1.07
O/C (mol/mol)	0.18	0.19	0.13	0.15	0.15	0.22
N/C (mol/mol)	0.03	0.03	0.06	0.06	0.06	0.01
HHV (MJ/kg)	34.31	33.36	35.16	33.29	33.22	31.03
ER (%)	47.48	53.43	41.12	32.45	40.42	40.77

^a^ Determined by difference, O = 100 − sum of (C, H, N, S, and Ash).

**Table 3 ijerph-19-10499-t003:** The effect of residence time on the characteristics of biocrude oils from the co-HTL of binary SS/CM.

Characteristics	BC-SS-30	BC-SS/CM-15	BC-SS/CM-30	BC-SS/CM-45	BC-SS/CM-60	BC-SS/CM-90	BC-SS/CM-120	BC-CM-30	Petro-Crude ^b^
C (wt%)	74.10	71.51	71.04	72.83	73.37	75.25	76.41	69.62	83–87
H (wt%)	10.22	9.16	8.54	7.566	7.998	8.94	9.12	7.01	10–14
O ^a^ (wt%)	10.73	16.23	16.64	15.46	14.71	11.75	10.75	21.55	0.1–2.0
N (wt%)	3.04	1.94	2.60	2.93	2.89	3.07	3.00	1.54	0.05–1.5
S (wt%)	1.91	1.16	1.18	1.219	1.037	1.00	0.72	0.29	–
H/C (mol/mol)	1.64	1.53	1.43	1.24	1.30	1.41	1.42	1.20	1.5–2.0
O/C (mol/mol)	0.11	0.17	0.18	0.16	0.15	0.12	0.11	0.23	<0.02
N/C (mol/mol)	0.04	0.02	0.03	0.03	0.03	0.03	0.03	0.02	<0.02
HHV (MJ/kg)	38.47	35.51	34.31	33.62	34.53	36.88	37.67	31.13	42–49
ER (%)	47.24	26.04	47.48	43.04	42.02	46.31	46.10	39.07	–

^a^ Determined by difference, O = 100 − sum of (C, H, N, S, and Ash). ^b^ Petro-crude data was collected from [40].

**Table 4 ijerph-19-10499-t004:** Elemental compositions, atomic ratios, and HHV of hydrochars from the co-HTL of binary SS/CM in light of residence time.

Characteristics	HC-SS-30	HC-SS/CM-15	HC-SS/CM-30	HC-SS/CM-90	HC-SS/CM-120	HC-CM-30
C (wt%)	11.05	32.14	21.50	20.55	17.87	40.94
H (wt%)	1.62	4.04	2.04	1.83	1.56	3.18
O ^a^ (wt%)	1.73	7.32	5.91	2.49	6.35	–
N (wt%)	1.03	1.15	1.90	1.45	1.28	2.01
S (wt%)	0.70	0.46	0.49	0.37	0.35	0.27
H/C (mol/mol)	1.75	1.5	1.13	1.06	1.04	0.92
O/C (mol/mol)	0.12	0.17	0.21	0.09	0.27	–
N/C (mol/mol)	0.08	0.03	0.08	0.06	0.06	0.04
HHV (MJ/kg)	5.89	15.88	9.44	9.29	7.53	18.45

^a^ Determined by difference, O = 100 − sum of (C, H, N, S, and Ash).

## Data Availability

Not applicable.

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
