# Peer review of "A Comprehensive Hydrothermal Co-Liquefaction of Diverse Biowastes for Energy-Dense Biocrude Production: Synergistic and Antagonistic Effects"

_ijerph, 2022, doi:10.3390/ijerph191710499_

Round 1
Reviewer 1 Report
This manuscript covers the information on biocrude production from sludge with diverse biowastes through HTL and co-HTL. In addition, the synergistic and antagonistic effects were studied through batch experiments. The finding of this study suggested that sewage sludge co-liquefied with other biowastes can serve as a double-edged solution for biowaste treatment and bioenergy production.
The manuscript is well written, and the figures and tables summarize the key components of the findings. However, some points must be addressed and discussed before the MS can be publishable in IJERPH.
· The study didn’t conduct mixing sludge (as one of the primary feedstock for HTL) with different biowastes; it was carried out by mixing four feedstocks (SS, CM, WS, and CP). In addition, the methodology of the study doesn’t align with the title of the manuscript (i.e., “of sludge with diverse biowastes”). If possible, I recommend changing the title to “of different diverse biowastes.”
· Ln 15, the 21.84wt% is for what? It is not clear in the abstract.
· Ln 93-95, the authors mentioned the sewage sludge (SS) was provided by Beipai water-quality testing center. The source of the SS is confusing with high ash content (Table 1) that needs to be verified and supported by the literature. Was the SS collected from the wastewater treatment plant?
· Ln 111-112, the authors selected the HTL temperature of 350°C from Reference 23, where the study primarily used glycerol as a co-solvent. The HTL reaction temperature should be chosen carefully for higher biocrude yield, not just from one study.
· In the method section, the biocrude recovery (Ln 128-130) needs to be verified; there is a significant chance of losing the lighter fraction of biocrude for air flowing at room temperature.
· Four feedstocks (SS, CM, WS, and CP) were considered in this study. However, they are no logical statement why the authors continued further with only SS and CM for different residence times for HTL.
· There is a significant impact on the quality of biocrude when co-HTL of different feedstocks. This study lacks the characterization of biocrude quality for quality assessment.
· In this study, the biocrude yields are comparatively lower than other recent studies from the literature. Since the HTL was conducted at 350°C, which was relatively high, it resulted in higher gas and volatiles fractions, as shown in Figures 2 (a and b). I recommend providing some comparisons with respect to other studies with HTL conditions (e.g., temperature, time, etc.)
· The legend of Figure 4 should include the HTL temperature of 350°C.
· The header of Table 2 is not clear/readable.
Reviewer 2 Report
The authors presented a very interesting paper on ssynergistic and antagonistic effects on biocrude yield and quality. The research and the presentation of the results were of a very high standard. However, the conclusions and lack of discussion were a bit disappointing. I would suggest after the conclusions, adding a paragraph on a brief discussion of the research results obtained.
There are minor shortcomings in the article which I have pointed out
- The degree index next to the temperature should be a superscript,
The equations are written in too large a font and are not aligned,
Chemical formulas have the sub-index written incorrectly,
Line 181 Please add an explanation of Yhc
Figure 2 marks (a) and (b) are too large for the whole figure.
Figure 6 should have a literature reference
The references list needs to be modified for the requirements of the journal
Round 2
Reviewer 1 Report
The authors have put considerable effort into the manuscript to address the comments/concerns. As a result, the paper is very much improved, and I have no problem recommending it for publication.